# The Sampling Method for Optimal Precursors of ENSO Event

Bin Shi[1,2] and Junjie Ma[3]

[1]Academy of Mathematics and Systems Science, Chinese Academy of Sciences, Beijing 100190, China
[2]School of Mathematical Sciences, University of Chinese Academy of Sciences, Beijing 100049, China
[3]School of Mathematics, North University of China, Taiyuan 030051, China

**Correspondence:** Bin Shi (Email: `shibin@lsec.cc.ac.cn`)

**Abstract.** El Niño-Southern Oscillation (ENSO) is one of the significant climate phenomena, which appears periodically in the tropical Pacific. The intermediate coupled ocean-atmosphere Zebiak-Cane (ZC) model is the first and classical one designed to numerically forecast the ENSO events. Traditionally, the conditional nonlinear optimal perturbation (CNOP) approach has been used to capture optimal precursors in practice. In this paper, based on state-of-the-art statistical machine learning techniques,[1] we investigate the sampling algorithm proposed in (Shi and Sun, 2023) to obtain optimal precursors via the CNOP approach in the ZC model. For the ZC model, or more generally, the numerical models with a large number $O(10^4 - 10^5)$ of degrees of freedom, the numerical performance, regardless of the statically spatial patterns and the dynamical nonlinear time evolution behaviors as well as the corresponding quantities and indices, shows the high efficiency of the sampling method by comparison with the traditional adjoint method. The sampling algorithm does not only reduce the gradient (first-order information) to the objective function value (zeroth-order information) but also avoids the use of the adjoint model, which is hard to develop in the coupled ocean-atmosphere models and the parameterization models. In addition, based on the key characteristic that the samples are independently and identically distributed, we can implement the sampling algorithm by parallel computation to shorten the computation time. Meanwhile, we also show in the numerical experiments that the important features of optimal precursors can be still captured even when the number of samples is reduced sharply.

## 1 Introduction

In the global climate system, the most prominent phenomenon of year-to-year fluctuations is El Niño–Southern Oscillation (ENSO), which makes a huge impact on Earth's ecosystems and human societies via influencing temperature and precipitation (Cashin et al., 2017). The natural interactions between ocean and atmosphere over the tropical Pacific not only alter weather around the world thus affecting marine and terrestrial ecosystems, such as fisheries, but also bring about secondary influences, such as human health and other societal and economic aspects of the Earth system (McPhaden et al., 2006; Tim-

---

[1]Generally, the statistical machine learning techniques refer to the marriage of traditional optimization methods and statistical methods, or, say, stochastic optimization methods, where the iterative behavior is governed by the distribution instead of the point due to the attention of noise. Here, the sampling algorithm used in this paper is to numerically implement the stochastic gradient descent method, which takes the sample average to obtain the inaccurate gradient.

mermann et al., 2018; Boucharel et al., 2021). Thus, it is of vital importance to learn the mechanism behind the set of coupled ocean-atmosphere phenomena in order to make a better forecast (Philander, 1989; Sarachik and Cane, 2010).

Perhaps the modern studies of the ENSO theory date back to the late sixties of the last century. Bjerknes (1969) pioneered the positive feedback mechanism, which explains why the ENSO system has two favored phases and their rapid growth. How-
ever, the positive feedback mechanism does not provide any explanation for why the ENSO event transits between the two phases. For the tropical atmosphere, Gill (1980) proposed a linear shallow water model on the equator with the first barocline mode vertical structure, which has become a standard tool used by both modelers and diagnosticians for describing the atmospheric response to the equatorially thermal forcing. Afterward, the heating parameterized in terms of sea surface temperature (SST) anomalies with arbitrary distribution was introduced in (Zebiak, 1982) and then convergence feedback parameteriza-
tion in (Zebiak, 1986). For the ocean process in the tropics, the governing equations including reduced gravity upper-ocean momentum equations and continuity equation for ocean thermocline depth were proposed in (Cane, 1984). Ultimately, Zebiak and Cane (1987) proposed the milestone coupled ocean-atmosphere model, Zebiak-Cane (ZC) model, to simulate the ENSO event, which imports the thermodynamical equation in terms of SST anomalies and couples the oceanic motion forced by the wind stress. Although the ZC model simulates the oscillation phenomenon, the mechanism still remains unclear in (Zebiak
and Cane, 1987). The so-called delayed oscillator theory proposed in (Suarez and Schopf, 1988; Battisti and Hirst, 1989) introduces the delayed negative feedback for the phase transition, where the core idea is the delayed effect of equatorial ocean waves. Based on the simulation of the ZC model, the well-known recharge-oscillator theory **was** first developed heuristically by Jin (1997a, b), where the key process is the zonal mean thermocline variation. Meanwhile, the ZC model is also the first intermediate coupled ocean-atmosphere numerical model used widely for the ENSO forecast. After it was proposed in (Zebiak
and Cane, 1987), there were many improvements in its predictability. The initialization procedure that incorporates the air-sea coupling was designed by Chen et al. (1995), which substantially improves the predictability of the ZC model. To predict the ENSO event, the ZC model was further improved by assimilating observed sea level data in (Chen et al., 1998). The LDEO5 version of the ZC model was exploited in (Chen et al., 2004), which successfully predicts all prominent El Niño events within the period 1857 to 2003 at lead times of up to two years.

In numerical prediction, a key issue that we often meet is the short-time behavior of a predictive model with imperfect initial data. In other words, it is of vital importance to understand the sensitivity of the numerical models to errors in the initial data. The simplest and most practical way is to estimate the likely uncertainty for the initial data polluted by the most dangerous errors. Currently, the conventional approach to capture the optimal initial perturbation is the so-called conditional nonlinear optimal perturbation (CNOP) approach innovatively introduced in (Mu et al., 2003), which is based on nonlinear optimization
methods. In the study (Duan et al., 2004), it was found that the CNOP approach using ZC-specified climatology with the seasonal cycle as the basic state produces optimal initial errors, which act as the optimal precursors for triggering ENSO events. Further investigation in (Mu et al., 2007) indicates that the optimal precursors are likely to contribute to the emergence of a significant spring predictability barrier (SPB). The SPB refers to a phenomenon in climate science where the predictability of systems, such as El Niño or La Niña, significantly decreases during the spring season. This is likely due to the transitional
nature of spring for ENSO, where signals are weak and noise is high, making predictions more challenging. Additionally, Duan

et al. (2008) recognized the decisive role of nonlinear temperature advection, and Yu et al. (2009) discovered two kinds of CNOP-type initial errors, a large-scale zonal dipolar pattern for the SST anomalies and a basin-side deepening or shoaling along the equator for the thermocline depth. The study by Mu et al. (2014) verified that the optimal precursors obtained in the ZC model exhibit significant similarity with the optimal initial growth errors, which are obtained by considering the ENSO events triggered the optimal precursors as a basic state. In addition, the ideas based on the CNOP approaches, or more general nonlinear optimization methods, have been generalized to rectify the model errors on the forecast of ENSO diversity in the ZC model, such as the SST cold-tongue cooling bias condition for the frequent occurrence of the central Pacific (CP)-type El Niño events in (Duan et al., 2014) and the nonlinear forcing singular vector (NFSV) perturbation that can distinguish the two kinds of El Niño events, the CP-type El Niño and the east Pacific (EP)-type El Niño in (Tao et al., 2020). Furthermore, an ensemble NFSV data assimilation approach is developed to address the ENSO forecast uncertainties caused by SPB and El Niño diversity (Zheng et al., 2023). Within the field of fluid mechanics, turbulence is widely regarded as a crucial and highly influential topic. The study of optimal energy growth was initially explored using the non-normal mode method in the seminal work by Reddy and Henningson (1993). Additionally, the scientific community has also developed the CNOP approach to investigate the disturbance of least amplitude for transition to turbulence. The CNOP approach, as described in (Pringle and Kerswell, 2010; Cherubini et al., 2010; Monokrousos et al., 2011), identifies the optimal precursors, referred to as minimal seeds, for the transition to turbulence. Nonlinear nonmodal analysis is applied to the 3D Navier-Stokes equation for an incompressible fluid to determine the optimal energy growth over all disturbances with a given starting energy and time horizon (Pringle and Kerswell, 2010; Cherubini et al., 2010; Monokrousos et al., 2011). Further details can be found in the comprehensive review (Kerswell, 2018) and an earlier review (Kerswell et al., 2014).

The CNOPs are often obtained by implementing nonlinear optimization methods, mainly including spectral projected gradient (SPG) method (Birgin et al., 2000), sequential quadratic programming (SQP) (Barclay et al., 1998), the limited memory Broyden-Fletcher-Goldfarb-Shanno (BFGS) algorithm (Liu and Nocedal, 1989) and the traditional method of Lagrange multipliers in practice.[2] As we know, the final state is the nonlinear evolution of the initial data polluted by some dangerous errors via a couple of nonlinear partial differential equations and some more complex parameterization models. Thus, the direct numerical computation of the gradient is so extremely expensive with the increase of degrees of freedom that it is unavailable in practice, since it needs to compute the Jacobian of the final reference state on the initial errors. The most popular and practical way to numerically approximate the gradient is the so-called adjoint technique, where the core is to exploit the adjoint model (Kalnay, 2003). Generally, the adjoint method reduces the computation time significantly at the cost of massive storage space to save the basic state. Even though a large amount of storage space has not been an essential issue based on the capabilities of modern computers, the adjoint model is still unusable for many numerical models, since the adjoint models

---

[2]It is worth noting that the first-order optimization method employed to obtain the maximum in the scientific community of fluid mechanics is the method of Lagrange multipliers (Kerswell, 2018), which has shown the consistent results when compared to other first-order optimization method mentioned in the following paragraph. The method of Lagrange multipliers is a classical method to solve the constrained optimization problem. It involves transforming the constrained optimization problem into an unconstrained one by incorporating the constrained condition into the Lagrange multipliers (Nocedal and Wright, 1999, Chapter 12). Additionally, the adjoint method is also explored to numerically compute the gradient. The details of the solution procedure can be found in (Kerswell, 2018, Section 3.2).

are hard to develop, especially for the coupled ocean-atmosphere models as well as the parameterization models (Wang et al., 2020). Based on state-of-the-art statistical machine learning techniques, Shi and Sun (2023) proposes the sampling algorithm to compute the CNOPs, which is prone to implementation in practice. Shi and Sun (2023) has successfully shown the efficiency of the sampling algorithm in the theoretical models, such as the Burgers equation with small viscosity and the Lorenz-96 model. Moreover, the computation time is shortened to the utmost at the cost of losing little accuracy. In this paper, we further implement the sampling algorithm to obtain the CNOPs in the realistic and predictive ZC model. Meanwhile, we show the efficiency of the sampling method by comparison with the adjoint method and discuss its available implementation in practice with modern parallel computation techniques. In addition, we also provide a positive answer for the open question of whether there exists an adjoint-free algorithm to obtain the CNOPs directly for the numerical models with a number $O(10^4 - 10^5)$ of degrees of freedom, which has already been listed in (Mu and Qiang, 2017; Kerswell, 2018; Wang et al., 2020).

The paper is organized as follows. Section 2 briefly describes how to numerically compute optimal precursors of the ENSO events in the ZC model, which includes the basic CNOP settings and the implementation of the sampling algorithm as well as how to carry it out by parallel computation in practice. The numerical performance of the sampling algorithm with the comparison of the adjoint method for the ZC model, in terms of the statically spatial patterns and the dynamical nonlinear time evolution behaviors as well as the corresponding quantities and indices, is shown in Section 3. Finally, we conclude this paper with a brief summary and discussion on some further research in Section 4.

## 2  Optimal Precursors via CNOP and Sampling

In this section, we first briefly describe the basic process to compute the optimal precursors by the use of the CNOP approach in the ZC model.[3] Then, based on the key characteristic that the samples are independently and identically distributed, we point out that the sampling method can be implemented efficiently by parallel computation and provide a detailed discussion.

### 2.1  The Basic CNOP Settings

Let $\mathscr{T} = (\mathscr{T}_{ij})$ and $\mathscr{H} = (\mathscr{H}_{ij})$ be SST anomalies and thermocline depth anomalies respectively,[4] where the index $i$ indicates the longitudinal grids in the region from 129.375 °E to 84.375 °W with the grid space 5.625° and the index $j$ indicates the latitudinal grids from 19 °S to 19 °N with the grid space 2°. From the classical references (Wang and Fang, 1996; Mu et al., 2007), we know that the characteristic scales of the SST anomalies and the thermocline depth anomalies are $|\mathscr{T}| \sim 2\,°C$ and $|\mathscr{H}| \sim 50\,m$, respectively. Then, the nondimensionalized quantities of the SST anomalies and the thermocline depth anomalies are given as

$$T = \frac{\mathscr{T}}{|\mathscr{T}|} = \frac{\mathscr{T}}{2°\mathbf{C}} \quad \text{and} \quad H = \frac{\mathscr{H}}{|\mathscr{H}|} = \frac{\mathscr{H}}{50\mathbf{m}}. \tag{1}$$

---

[3]Although the CNOP approach has been extended to investigate the influences of boundary errors and model errors on atmospheric and oceanic models (Wang et al., 2020), here we only explore the impact of initial errors.

[4]Throughout the paper, all the vectors are denoted by the bold italics.

Moreover, in the ZC model, the dominant factors that influence the ENSO events are the SST anomalies and the thermocline depth anomalies (Zebiak and Cane, 1987). With (1), the initial errors that we need to consider should include these two variables as $\boldsymbol{u}_0 = (\boldsymbol{T}(0), \boldsymbol{H}(0))$. For the quantity used to measure, we adopt the standard Euclidean norm as

$$\|\boldsymbol{u}_0\| = \|(\boldsymbol{T}(0), \boldsymbol{H}(0))\| = \sqrt{\sum_{i,j} \left[ T(0)_{ij}^2 + H(0)_{ij}^2 \right]}. \tag{2}$$

Next, we consider the objective function that is on the initial errors. As our primary concern is maximizing the target quantity solely dependent on the nonlinear evolution state of the SST anomalies, we define the objective function as

$$J(\boldsymbol{u}_0) = \|\boldsymbol{T}(\boldsymbol{u}_0, \tau)\|^2, \tag{3}$$

where $\|\cdot\|$ is still the Euclidean norm and $\tau$ is the prediction time set as 9 months in this paper. With (2) and (3), we derive the constrained nonlinear optimization problems for the optimal precursors, that is, the CNOPs in the ZC model as

$$\max_{\|\boldsymbol{u}_0\| \le \delta} J(\boldsymbol{u}_0) \tag{4}$$

where the constraint parameter is set as $\delta = 1.0$.

## 2.2 The Sampling Method and Parallel Computation

Based on Stokes' formula, Shi and Sun (2023) proposes the sampling algorithm, which reduces the gradient to the function value in the sense of expectation. Simply speaking, we consider the average of the function values in a small ball instead of the exact function value. The rigorous representation is to take the expectation of the function values along the following way as

$$\hat{J}(\boldsymbol{u}_0) = \mathbb{E}_{\boldsymbol{v}_0 \in \mathbb{B}^d} \left[ \nabla J(\boldsymbol{u}_0 + \epsilon \boldsymbol{v}_0) \right] \tag{5}$$

where $\mathbb{B}^d$ is the unit ball in $\mathbb{R}^d$ and $\epsilon > 0$ is a small real number. According to Stokes' formula, we can derive the gradient of the expectation (5) as

$$\nabla \hat{J}(\boldsymbol{u}_0) = \mathbb{E}_{\boldsymbol{v}_0 \in \mathbb{B}^d} \left[ \nabla J(\boldsymbol{u}_0 + \epsilon \boldsymbol{v}_0) \right] = \frac{d}{\epsilon} \cdot \mathbb{E}_{\boldsymbol{v}_0 \in \mathbb{S}^{d-1}} \left[ J(\boldsymbol{u}_0 + \epsilon \boldsymbol{v}_0) \boldsymbol{v}_0 \right], \tag{6}$$

where $\mathbb{S}^{d-1}$ is the $(d-1)$-dimensional unit sphere. Following the expression (6), we can take the sample average to numerically approximate the gradient as

$$\nabla \hat{J}(\boldsymbol{u}_0) \approx \frac{d}{n\epsilon} \sum_{i=1}^{n} \left[ J(\boldsymbol{u}_0 + \epsilon \boldsymbol{v}_{0,i}) \boldsymbol{v}_{0,i} \right], \tag{7}$$

where $n$ is the number of samples and $\boldsymbol{v}_{0,i}, (i = 1, \ldots, n)$ are the random variables identically sampled from the uniform distribution on the unit sphere $\mathbb{S}^{d-1}$. By utilizing the sample average of function values (7) as an approximate gradient, we can employ various gradient accent methods within the constraint domain, such as SPG, SQP, BFGS and the method of Lagrange multiplier, which help us maximize the objective function $J(\boldsymbol{u_0})$. In this paper, the specific gradient method within the

constraint domain that we utilize is the second spectral projected gradient (SPG2) method mentioned, as mentioned in (Birgin et al., 2000). The rigorous Chernoff-type bound for the sample average with the exact gradient has been derived in (Shi and Sun, 2023, Section 3 and Appendix A).

The average of function values (7) indicates that the random variables $\boldsymbol{v}_{0,i}, (i = 1, \ldots, n)$ are independently sampled from the uniform distribution on the unit sphere $\mathbb{S}^{d-1}$. This means that for any two samples, $\boldsymbol{v}_{0,i}$ and $\boldsymbol{v}_{0,j}$, where the indices $i$ and $j$ satisfy $i \neq j$, there is no relationship between them. In other words, every sample $\boldsymbol{v}_{0,i}, (i \in \{1, \ldots, n\})$ has no influence with each other and is drawn independently. With modern parallel computation techniques, it is possible to run the numerical model and obtain the values $J(\boldsymbol{u}_0 + \epsilon \boldsymbol{v}_{0,i})\boldsymbol{v}_{0,i}$ for each $i \in \{1, \ldots, n\}$ simultaneously, assuming unlimited computational resources are available. This parallelization allows for efficient computation and reduces the time required to run $n$ instances of the numerical model to that of running the model only once. However, it is important to note that in the adjoint method, the process of running the numerical model involves two consecutive steps. Initially, there is a forward numerical integration from 0 to $\tau$, followed by a backward numerical integration from $\tau$ to 0. These computations are based on the data obtained by running the numerical model. This process is executed in a single-thread manner, meaning that parallel computation is not applicable. On the other hand, in the implementation of the sampling algorithm, the process of running the numerical model only requires forward numerical integration from 0 to $\tau$, without any backward numerical integration. This implies that for each sample, we only need to run the forward numerical integration once. Since the samples are independent, we can leverage the parallel computation to implement the sampling algorithm, which further reduces the time required for running the forward numerical integration once. With the current resource of computation, we have successfully implemented the sampling algorithm to obtain the CNOPs, or the optimal precursors of the ZC model. This numerical model has a substantial number of degrees of freedom, estimated to be on the order of $O(10^4 - 10^5)$. The implementation utilizes the modern parallel computation technique. The numerical performance, including the spatial patterns, objective values, computation times and nonlinear evolution of Nino 3.4 index, is shown in Section 3.

## 3 The Numerical Performance

After the CNOP approach is imported to the ZC model (Mu et al., 2007), the adjoint method has always been the baseline algorithm in practice. In this section, we show the numerical performance of the sampling algorithm by comparison with the adjoint method in the ZC model. The static spatial patterns of the optimal precursors with some measurement quantities and computation times are shown in Section 3.1, while the nonlinear time evolution behaviors of the optimal precursors and the corresponding Niño 3.4 SST anomaly index in Section 3.2

### 3.1 The optimal precursors in the ZC model

Recall the optimal precursors bringing about the El Niño event, which is obtained by the CNOP approach in (Yu et al., 2009). The spatial pattern in terms of SST anomalies is manifested as a large-scale zonal dipolar pattern, the warm pole of about $0.2°C$ along the equator in the east Pacific and the cold one of about $0.2°C$ in the central Pacific; while a basin-side deepening

about 100m along the equator is the character of that for thermocline depth anomalies. We reproduce the spatial patterns of the optimal precursors bringing about the El Niño event in Figure 1, the two pictures in the top row. When we take 1000 samples to

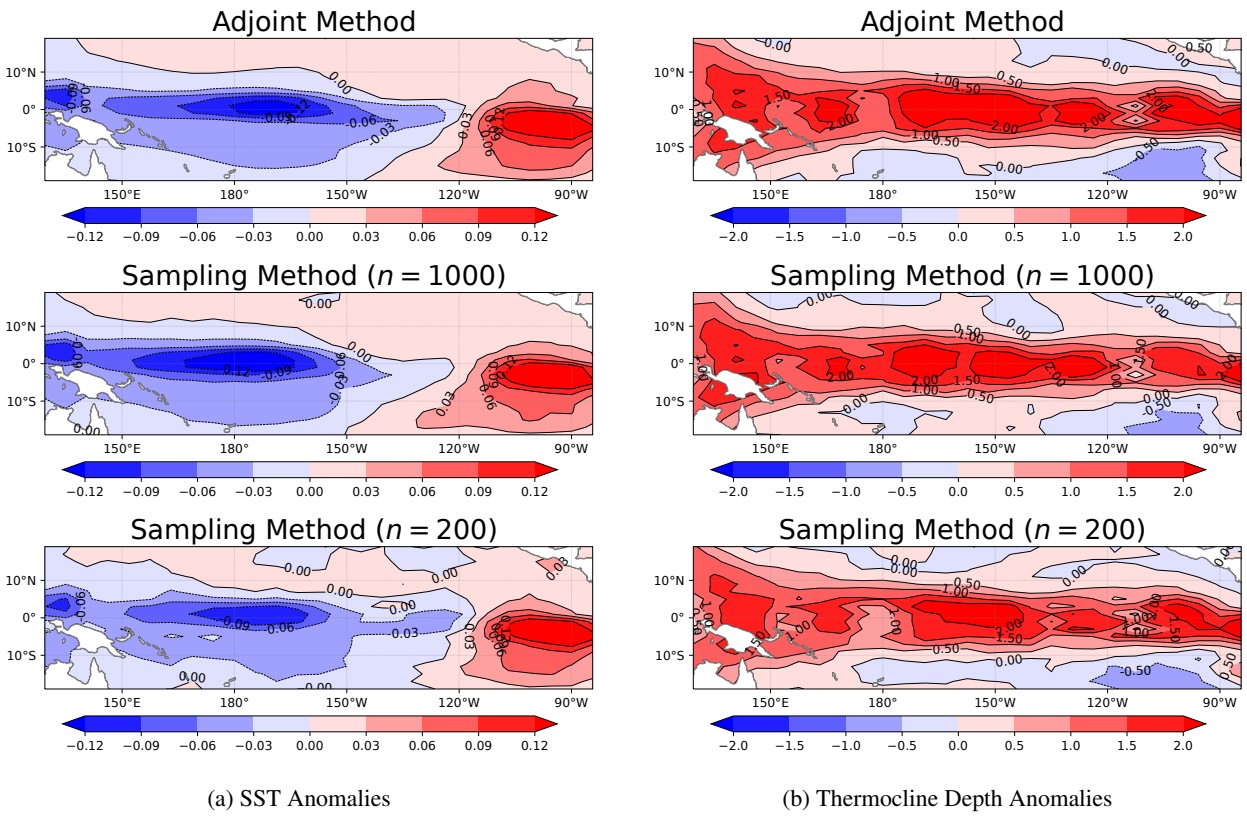

(a) SST Anomalies

(b) Thermocline Depth Anomalies

**Figure 1.** The spatial patterns of the optimal precursors in terms of SST anomalies (Left Column) and thermocline depth anomalies (Right Column). The prediction time is 9 months. By the rows from top to bottom, the spatial patterns are obtained by the adjoint method, the sampling method with $n = 1000$ and that with $n = 200$, respectively.

implement the sampling method, both the spatial patterns of the optimal precursors in terms of SST anomalies and thermocline
depth anomalies are almost identical to that obtained by the adjoint method, which is shown in the middle row of Figure 1.
Furthermore, when the number of samples is reduced from 1000 to 200, we can find from the two pictures in the bottom row
of Figure 1 that both the large-scale zonal dipolar pattern of SST anomalies and the basin-side deepening pattern of thermocline
depth anomalies for the optimal precursors leading to the El Niño event can be still captured, even though there are some small
deviations due to some noise.

We have shown considerable similarities in the spatial patterns of the optimal precursors bringing about the El Niño event
in Figure 1, which are obtained by the adjoint method, the sampling method with $n = 1000$ and that with $n = 200$. If these
can be viewed as qualitative similarities, we still need to verify the similarities of the optimal precursors from these numerical
algorithms quantitatively. The objective values $J(\boldsymbol{u}_0)$ obtained using the optimal precursors $\boldsymbol{u}_0$ computed by both the adjoint

method and the sampling method are shown in Table 1, where we can find that the value computed by the adjoint method

| Methods Objective | Adjoint | Sampling ($n = 1000$) | Sampling ($n = 200$) |
|---|---|---|---|
| Values ($J(\boldsymbol{u}_0) = \|\boldsymbol{T}(\tau)\|^2$) | 16.8441 | 16.6307 | **15.4193** |
| Percentage | 100% | 98.73% | **91.54%** |

**Table 1.** The optimal precursors $\boldsymbol{u}_0$ are computed by both the adjoint method and the sampling method with the corresponding spatial patterns shown in Figure 1. First Line: The objective values $J(\boldsymbol{u}_0)$ computed in eq. (4). Second Line: The percentages over the values computed by the adjoint method. Bold highlights the high efficiency of the sampling method with $n = 200$.

is 16.8441 and the values by the sampling method with $n = 1000$ and $n = 200$ are 16.6307 and 15.4193, respectively. The objective values obtained by the sampling algorithm look very close to the one of the baseline adjoint method. Here, we can further show the similarities by taking the ratio between them. If the objective value obtained by the adjoint method is taken as the numerator, we can find that the objective value obtained by the sampling method with $n = 1000$ takes the percentage 98.73%, which manifests the objective values obtained by the two algorithms are almost identical. When the number of samples 190 is reduced from 1000 to 200, the percentage that the objective value obtained by the sampling algorithm occupies decreases to 91.54%, which is still more than 90% and shows quite high similarities.

Both the spatial patterns and the objective values indicate that the optimal precursor obtained through the sampling algorithm, using only 200 samples, is very similar to the one obtained through the baseline adjoint algorithm. To show the efficiency of the sampling method, a comparison of computation times is necessary. As mentioned in Section 2.2, the sampling 195 algorithm, implemented with parallel computation, reduces the computation of the gradient by performing a single forward numerical integration. In contrast, the adjoint method requires a two-step process involving both forward and backward numerical integrations. We have realized them and recorded the computation times of both the adjoint method and the sampling algorithm implemented with parallel computation in Table 2. It needs to take about 50 iterations by implementing both the

| Methods | Adjoint | Sampling (Parallel Computation) |
|---|---|---|
| Computation Time (50 Iterations) | $\approx 15s$ | $\approx 3s$ |
| Computation Time per Iteration | $\approx 0.3s$ | $\approx 0.06s$ |

**Table 2.** The comparison of computation times between the adjoint method and the sampling method under the parallel computation. Run the Fortrun code on the CPU: Intel® Xeon® Gold 6132 Processor, 19.25M Cache, 2.60 GHz with 8 nodes and 28 cores per node.

adjoint method and the sampling method to get the optimal precursors. On the supercomputer, the adjoint method takes about 200 $15s$, that is, about $0.3s$ per iteration; while the sampling method takes about $3s$ when the implementation is under the parallel computation, that is, about $0.06s$ per iteration. Furthermore, when the sampling method is implemented, we avoid running the numerical model reversely such that the computation time is shortened to 1/5. Without any doubt, the computation that is reduced must be implemented by parallel computation. However, based on the current resource of computation, it is available

for us to implement the sampling method under the parallel computation to obtain the optimal precursors by the use of the

205 CNOP approach in the ZC model, more generally, the numerical model with a number $O(10^4 - 10^5)$ of degrees of freedom.

## 3.2 The nonlinear time evolution behavior of the optimal precursors

Based on the CNOP approach, the statically spatial patterns of the optimal precursors of ENSO events are in terms of both SST anomalies and thermocline depth anomalies. In Section 3.1, we have shown the high efficiency of the sampling algorithm by the comparison with that obtained by the baseline adjoint method as well as the computation times. However, we still need

to study the dynamic behaviors of the ENSO events to predict the potential impacts, where a great way is to only monitor the nonlinear evolution of SST anomalies.

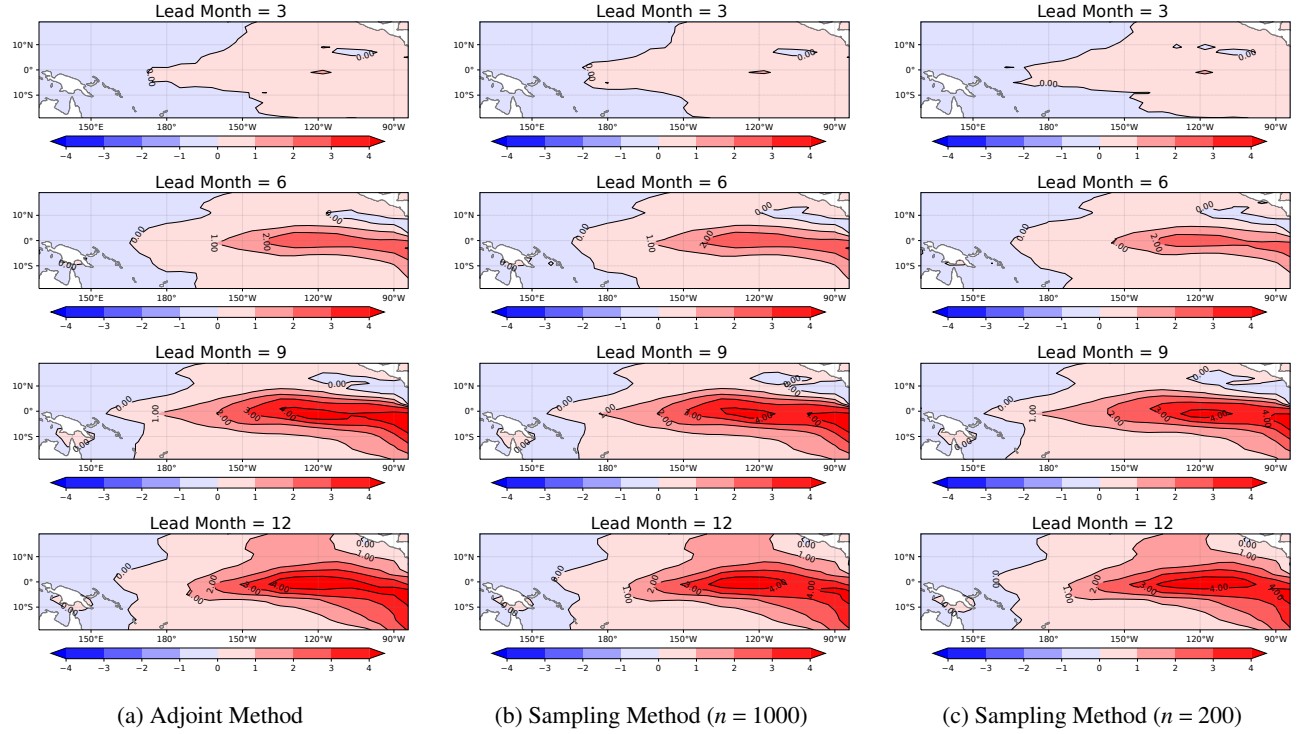

(a) Adjoint Method       (b) Sampling Method ($n = 1000$)       (c) Sampling Method ($n = 200$)

**Figure 2.** The spatial patterns of the nonlinear time evolution of the optimal precursors in terms of SST anomalies.

Recall the nonlinear time evolution of SST anomalies simulated by the coupled ocean-atmosphere ZC model shown in (Yu et al., 2009), where the optimal precursor, or the CNOP, is obtained by the adjoint method. By adding the initial optimal precursors to the climatological mean equilibrium state, we run the ZC model to reproduce the EP-type El Niño phenomenon

in the left column of Figure 2, where we can observe that the warm phase in the east Pacific along the equator is intensified gradually with the season evolution in one year, that is, the lead time is set as 3, 6, 9, and 12 months, respectively. More concretely, in the east Pacific along the equator, the region of the warm phase is gradually enlarged and the SST anomalies are raised up sharply from about 0°C to 8°C. In the right two columns of Figure 2, we show the spatial patterns in terms of the

nonlinear time evolution of SST anomalies, where the initial condition starts from the climatological mean equilibrium state
added by the initial optimal precursors obtained by the sampling methods with $n = 1000$ and $n = 200$. By taking a comparison
between the spatial patterns shown from the left to the right in Figure 2, the nonlinear time evolution behaviors of the initial
optimal precursors are also remarkably similar to each other with the change of seasons. Even though the number of samples
is reduced to 200, we can still find that the spatial patterns in terms of the seasonal evolution of SST anomalies are almost
consistent with the baseline one starting from initial optimal precursors obtained by the adjoint method.

Based on the nonlinear time evolution of SST anomalies simulated in Figure 2, we have shown qualitatively the similarities
of the dynamical behaviors of the initial optimal precursors obtained by both the baseline adjoint method and the sampling
method. Nevertheless, we still need to show the similarities quantitatively for the dynamical evolution of SST anomalies from
the three kinds of initial optimal precursors. Currently, the main variable that is considered from the ENSO forecasts of the

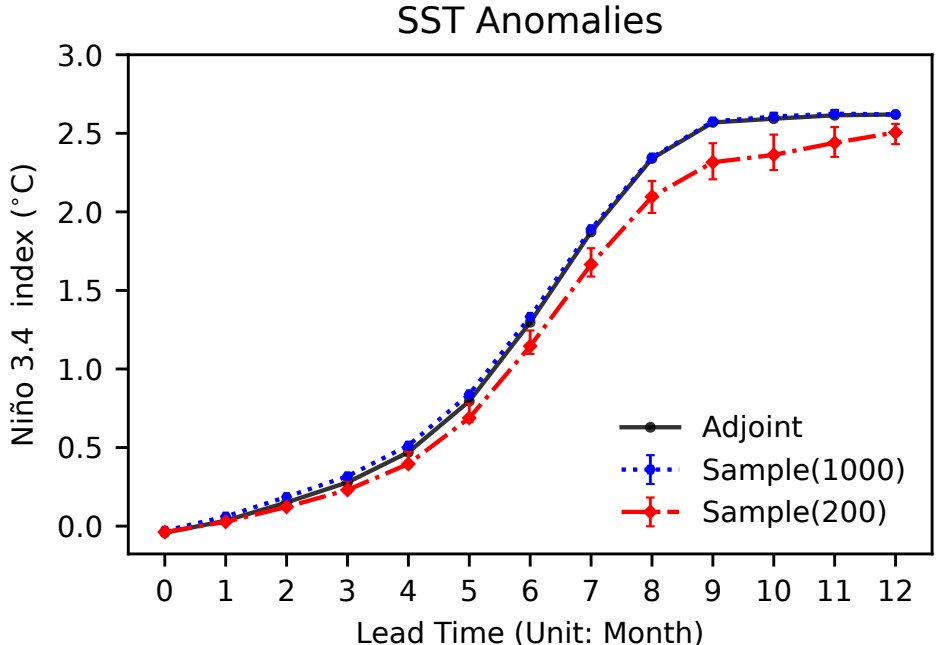

**Figure 3.** The nonlinear time evolution of Niño 3.4 SST anomaly index within a model year. The bars represent the range of errors obtained from running the sampling method 50 times.

coupled climate models is the Niño 3.4 SST anomaly index, which is used by the National Climate Centre (NCC) in Australia to
classify ENSO conditions. Here, we show that the Niño 3.4 SST anomaly indices change nonlinearly along the time evolution
line within a model year in Figure 3, where the three dynamical curves generated by these proposed algorithms are quite close
to each other. Furthermore, we can observe in Figure 3 that the dynamical curve of the Niño 3.4 SST anomaly index starting
from the initial optimal precursor obtained by the sampling method with $n = 1000$ almost coincides with the baseline one from

the initial optimal precursor obtained by the adjoint method. When the number of samples is reduced from 1000 to 200, some small deviations appear in the dynamical curve of the Niño 3.4 SST anomaly index. Thus, it is necessary for us to quantify these derivations such that we can study the accuracy of the Niño 3.4 SST anomaly index by implementing the sampling algorithm to approximate that generated by the adjoint method when the number of samples is reduced from 1000 to 200. Taking the Niño 3.4 SST anomaly index generated by the adjoint method as a basis, we compute the relative Niño 3.4 SST anomaly index, that is, the difference of the Niño 3.4 SST anomaly indices from between the baseline adjoint method and the sampling method in Figure 4. Here, we can find that the relative Niño 3.4 SST anomaly index from the sampling method with $n = 1000$ takes

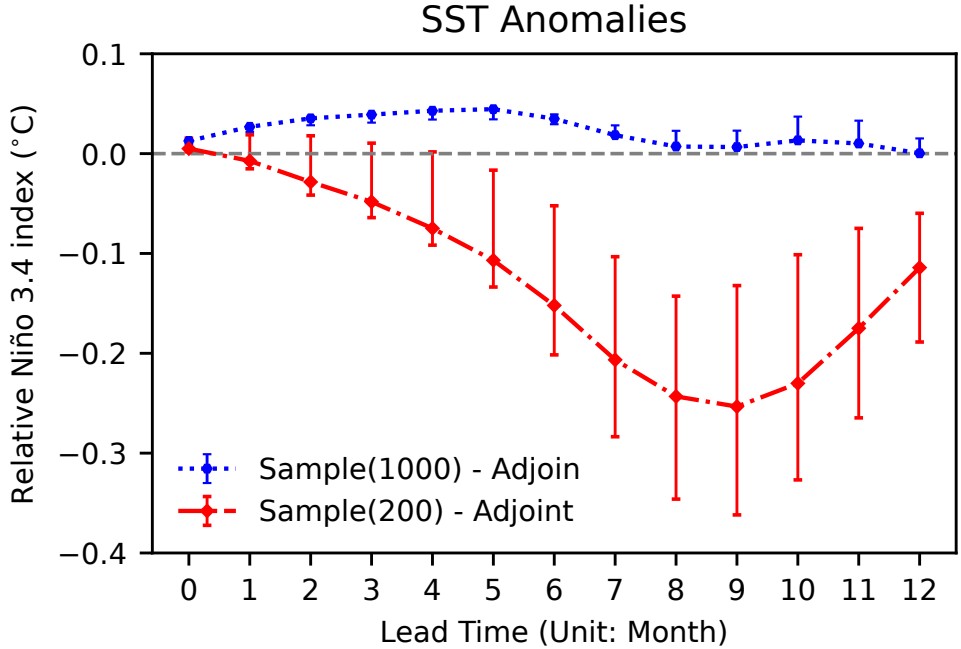

**Figure 4.** The nonlinear time evolution of relative Niño 3.4 SST anomaly index within a model year. (the Niño 3.4 SST anomaly index obtained by the sampling method minus that by the baseline adjoint method) The bars represent the range of errors obtained from running the sampling method 50 times.

the characteristic scale with $O(10^{-2})$, while that from the sampling method with $n = 200$ is $O(10^{-1})$. In other words, when we implement the sampling method by reducing the number of samples from $n = 1000$ to $n = 200$, the relative Niño 3.4 SST anomaly index is degraded from $O(10^{-2})$ to $O(10^{-1})$, which quantitatively manifests that the accuracy of the Niño 3.4 SST anomaly index is loosened up to an order of magnitude. However, if we take the comparison with the Niño 3.4 SST anomaly index, whose characteristic scale is $O(1)$, the numerical errors by reducing the number of samples from 1000 to 200 are still too small to influence the nonlinear time evolution of SST anomalies.

## 4 Summary and discussion

Based on the state-of-the-art statistical machine learning techniques, the sampling method to compute the CNOPs is proposed in Shi and Sun (2023). In this paper, we successfully implement the sampling method to obtain the initial optimal precursors in the realistic and predictive ZC model, more generally, the numerical model with a number $O(10^4 - 10^5)$ of degrees of freedom. The sampling method with fewer samples can achieve consistent performance as the adjoint method in the numerical experiments, regardless of the statically spatial patterns and the dynamical nonlinear time evolution behaviors as well as the corresponding quantities and indices. By leveraging the key characteristic that the samples are independently and identically distributed, we can effectively implement the sampling method using the modern parallel computation technique. This approach eliminates the need to run the numerical model in reverse, leading to a significant reduction in computation time. In fact, the computation time can be shortened by approximately 1/5, allowing for more efficient and faster processing. In general, the number of samples required for a numerical model depends on the degrees of freedom. As the degrees of freedom increase, a larger number of samples is typically needed. However, the nonlinear evolution of the initial values within the numerical model itself should not be overlooked. This has been empirically demonstrated in a comparison between the Burgers equation and the Lorenz-96 model (Shi and Sun, 2023). In the case of the Burgers equation, which exhibits weak nonlinear evolution, achieving the desired experimental effect can be accomplished with just 5 samples, even with 100 degrees of freedom. On the other hand, the 40-dimensional Lorenz-96 model, characterized by strong nonlinear evolution, also requires 5 samples to achieve the desired effect. Based on empirical observations, a good strategy for initial experiments is to choose the number of samples to be approximately equal to the square root of the number of degrees of freedom, that is, $n \approx \sqrt{d}$. Indeed, by implementing the sampling algorithm with 60 samples, we are able to achieve a numerical performance that nearly reproduces the results obtained by the baseline adjoint method for the optimal precursors. However, it has been observed that the numerical results are unstable. Out of four runs, only one consistently produces correct numerical performance. Besides, by the use of the CNOP approach in the coupled ocean-atmosphere ZC model, we can obtain the other kind of optimal precursors, which leads to the La Niña event. However, due to the deficiency of the original ZC model in (Zebiak and Cane, 1987), a warm tendency of the Niño 3.4 SST anomaly index will appear after it decreases to the coldest point for the La Niña event, which is shown in (Duan et al., 2008). In our numerical experiments, the numerical performance based on the optimal precursors leading to the La Niña event can be also obtained. Thus, these numerical experiments are not representative **so** that we neglect to show their numerical performance in the paper.

For a realistic global climate system model (GSCM) or atmosphere-ocean general circulation model (AOGCM), it is often impractical to develop the adjoint model, so the sampling method provides a probable way of computing the CNOPs to investigate its predictability. An interesting direction for further research is to investigate the CNOPs computed by the sampling method in the numerical models that are used in realistic prediction and forecast, such as the Weather Research and Forecasting (WRF) Model, a state-of-the-art mesoscale numerical weather prediction system for operational forecasting applications. Another interesting direction is to attempt to use the sampling method to realize a more (or less) nonlinearly stable flow by changing some aspect of the system (Cherubini and De Palma, 2013; Rabin et al., 2014; Passaggia and Ehrenstein, 2013),

where the adjoint technique still needs to take costly optimization calculations (Kerswell, 2018). In addition, the traditional data assimilation is based on the development of the adjoint model (Kalnay, 2003). In this paper, our numerical experiments make the four-dimensional variational (4D-Var) data assimilation technique become available possibly on the coupled climate system models as well as the parameterization models. Therefore, it is valuable and thrilling to implement the sampling method to process 4D-Var data assimilation in realistic systems, such as the Flexible Global Ocean-Atmosphere-Land System (FGOALS)-s2 (Wu et al., 2018) for decadal climate prediction. Finally, we conclude this paper with a statement that the sampling method, based on state-of-the-art machine learning techniques, is a probable way to realize the nonlinear optimization method in practice to address these challenges in (Kerswell, 2018; Wang et al., 2020).

*Code and data availability.* The datasets generated and/or analyzed during the study are stored on computers at the State Key Laboratory of Numerical Modeling for Atmospheric Sciences and Geophysical Fluid Dynamics (LASG) and will be available to researchers upon request.

*Author contributions.* Bin Shi constructed the basic idea of this paper, derived all formulas, and wrote the paper. Junjie Ma coded the sampling method in the ZC model in Fortran (Figures drawn by Python) and joined the discussions of this paper. Both authors contributed to the writing of the paper.

*Competing interests.* The authors declare that they have no conflict of interest.

*Acknowledgements.* Junjie Ma joins in this work during the final semester of his Ph.D. under the supervision of Wansuo Duan at the Institute of Atmospheric Physics, Chinese Academy of Sciences. This work was supported by Grant No.12241105 of NSFC and Grant No.YSBR-034 of CAS.

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
