# Peer review of "The Sampling Method for Optimal Precursors of ENSO Event"

_EGUsphere, 2023_

## Referee Comment (RC2)

**Review of: *"The Sampling Method for Optimal Precursors of ENSO Event"**

June 2023

**Overview**

An algorithm for computing optimal precursors for ENSO events using the Zebiak-Cane model. Such algorithm consists of independently sampling the dynamical model in order to obtain the mode that maximizes the amplification of SST anomaly (for El Niño events). The maximization is carried out by the theory previously published in Shi and Sun 2023, in Nonlinear Processes in Geophysics. Their results contain the spatial structure of said precursors and their ability to describe ENSO events is tested against the adjoint method benchmark.

In my opinion, the article is concise, well written and the topic matches the journal's scope. The results presented herein are novel, since they stem from an application of a recently published theory in this very journal. I would recommend its publication, provided that my comments are considered.

**General comments**

1. In the Introduction (Lines 50-55), the authors state that "there is a great similarity between optimal precursors and the optimal initial errors obtained by CNOP approaches, in terms of spatial structure and localization." The readers would appreciate a few sentences as to why this is the case, since it is not, in my opinion, trivial. Are the optimal initial errors the predictors for ENSO in the ZC model?

2. In Lines 64-70, the authors speak about optimal energy growth. Typically optimal energy is attained when the system (say, Navier-Stokes) is initialized with the optimal mode. Can the authors say something about the formal/mathematical connection between these optimal modes and the adjoint methods? Could they potentially have the same spatial structure as other CNOP approaches? Is there literature that applies the optimal growth theory with ENSO?

3. Regarding the optimal energy growth, it might be useful to cite the seminal work on linear energy growth by Reddy and Henningson:

   *Reddy, S., and Henningson, D. (1993). Energy growth in viscous channel flows. Journal of Fluid Mechanics, 252, 209-238. doi:10.1017/S0022112093003738*

4. Line 114: can the authors explain the choice of $\tau = 9$?

5. Line 114: can the CNOP's spatial structure change as a function of $\tau$?

6. I understand the Section 2.2 is a brief summary of Shi and Sun 2023, although the reader would benefit from a couple extra words on the actual statistical machine learning algorithm employed. How is the objective function actually maximised? If evaluating $J(\mathbf{u}_0 + \epsilon \mathbf{v}_0)$ requires solving the equations to find out $\mathbf{T}(\tau)$, how can it be less costly than the adjoint method? Can the authors give an intuition?

7. Line 235: For La Niña events, how would your algorithm change (in the language of Section 2.2)?

8. For Figures 3 and 4, some sort of error bars would be useful. Could the authors possibly use the theory and bounds of Shi and Sun 2023 to compute some uncertainty bars?

**Minor/Technical comments**

1. Line 2: I suggest using "tropical Pacific" instead of "tropic Pacific".

2. Line 51: It would be helpful to clarify what "spring predictability barrier" means and its causes.

3. Line 60: I suggest to use SPB instead of "spring predictability barrier".

4. Line 61: remove "the" in "the turbulence".

5. Line 67: replace "reivew" by "review".

6. Line 94: replace "futhre" by "further".

7. Lines 111-112: awkward phrasing: "Since the target quantity required to maximize that we concern...".

8. Eq. (6): "$\mathbb{S}^{d-1}$" is not defined until way later. I would suggest defining it immediately below.

9. Line 237: replace "such" by "so".

---

## Author Response (AR1)

**The Response to Reviewers' Comments**

We thank the editor and the two referees for thoroughly reading the manuscript and their helpful comments. We are very pleased to see many positive remarks. For example, the first reviewer said, "This is an interesting paper as it demonstrates the practical application of the algorithm proposed by one of the authors to real atmospheric dynamic models." Reviewer 2 added, "The results presented herein are novel, since they stem from an application of a recently published theory in this very journal." In light of the comments, we have made a thorough revision addressing all major concerns, resulting in a significantly improved paper version.

**1   The First Reviewer's Report**

**1.1   Comments**

1. *In equation (1), I found it more intuitive to keep the unit in the dinominantor if the goal is to nondimensionalize the variable.*

   Thanks for the good comment. We have revised it as

   $$\boldsymbol{T} = \frac{\mathscr{T}}{|\mathscr{T}|} = \frac{\mathscr{T}}{2°\mathbf{C}} \quad \text{and} \quad \boldsymbol{H} = \frac{\mathscr{H}}{|\mathscr{H}|} = \frac{\mathscr{H}}{50\mathbf{m}}.$$

2. *Table 1: to make it more clear, the authors need to explicitly say that the values reported in table are the original objective values formulated in (4) instead of the sample average (5).*

   Thanks, we have revised the caption in Table 1 as "**The optimal precursors $\boldsymbol{u}_0$ are computed by both the adjoint method and the sampling method with the corresponding spatial patterns shown in Figure 1. First Line: The objective values $J(\boldsymbol{u}_0)$ computed in eq.(4). Second Line: The percentages over the values computed by the adjoint method.**". We have also revised the sentence in Lines 186-187 as " The objective values $J(\boldsymbol{u}_0)$ **obtained using the optimal precursors $\boldsymbol{u}_0$ computed by both the adjoint method and the sampling method** are shown in Table 1."

3. *Table 2: more discussions are needed for the settings of the experiment. For example how many cores are used in parallel sampling? Is multithreading allowed for the adjoint method?*

   Thanks for the comments. In parallel sampling, we use 200 cores in the supercomputer with 8 nodes and 28 cores per node, which is labeled in the caption of Table 2. The multithreading is not useful for the adjoint method. Following the suggestion, we have revised the sentences of the whole paragraph in Line 143 – Line 162 as "**The average of function values (7) indicates that the random variables $\boldsymbol{v}_{0,i}, (i = 1, \ldots, n)$ are independently sampled from the uniform distribution on the unit sphere $\mathbb{S}^{d-1}$. This means that for any two samples, $\boldsymbol{v}_{0,i}$ and $\boldsymbol{v}_{0,j}$, where the indices $i$ and $j$ satisfy $i \neq j$, there is no relationship between them. In other words, every sample $\boldsymbol{v}_{0,i}, (i \in \{1, \ldots, n\})$ has no influence with each other and is drawn independently. With modern parallel computation techniques, it is possible to run the numerical model and obtain the**

values $J(\boldsymbol{u}_0 + \epsilon\boldsymbol{v}_{0,i})\boldsymbol{v}_{0,i}$ for each $i \in \{1, \ldots, n\}$ simultaneously, assuming unlimited computational resources are available. This parallelization allows for efficient computation and reduces the time required to run $n$ instances of the numerical model to that of running the model only once. However, it is important to note that in the adjoint method, the process of running the numerical model involves two consecutive steps. Initially, there is a forward numerical integration from $0$ to $\tau$, followed by a backward numerical integration from $\tau$ to $0$. These computations are based on the data obtained by running the numerical model. This process is executed in a single-thread manner, meaning that parallel computation is not applicable. On the other hand, in the implementation of the sampling algorithm, the process of running the numerical model only requires forward numerical integration from $0$ to $\tau$, without any backward numerical integration. This implies that for each sample, we only need to run the forward numerical integration once. Since the samples are independent, we can leverage the parallel computation to implement the sampling algorithm, which further reduces the time required for running the forward numerical integration once. With the current resource of computation, we have successfully implemented the sampling algorithm to obtain the CNOPs, or the optimal precursors of the ZC model. This numerical model has a substantial number of degrees of freedom, estimated to be on the order of $\mathrm{O}(10^4 - 10^5)$. The implementation utilizes the modern parallel computation technique. The numerical performance, including the spatial patterns, objective values, computation times and nonlinear evolution of Nino 3.4 index, is shown in Section 3." and added more sentences in Line 194 – Line 199 as **To show the efficiency of the sampling method, a comparison of computation times is necessary. As mentioned in Section 2.2, the sampling algorithm, implemented with parallel computation, reduces the computation of the gradient by performing a single forward numerical integration. In contrast, the adjoint method requires a two-step process involving both forward and backward numerical integrations. We have realized them and recorded the computation times of both the adjoint method and the sampling algorithm implemented with parallel computation in Table 2.**

4. *The original paper [Shi and Sun, 2023] applied the algorithm to some traditional dynamic system toy models such as viscous Burger's equation and Lorenz 96 model, and it only requires 5-15 samples for the method to perform well in these low-dimensional settings. It would be great if the authors could include some discussions about convergence results and how the number of samples scales with the dimensionality of the models.*

Thanks for the good comment. We have revised and added more sentences in Line 258 – Line 270 to discuss the convergence results and empirically how to scale the number of samples with the dimensionality of the models as " **In general, the number of samples required for a numerical model depends on the degrees of freedom. As the degrees of freedom increase, a larger number of samples is typically needed. However, the nonlinear evolution of the initial values within the numerical model itself should not be overlooked. This has been empirically demonstrated in a comparison between the Burgers equation and the Lorenz-96 model (Shi and Sun, 2023). In the case of the Burgers equation, which exhibits weak nonlinear evolution, achieving**

the desired experimental effect can be accomplished with just $5$ samples, even with $100$ degrees of freedom. On the other hand, the $40$-dimensional Lorenz-96 model, characterized by strong nonlinear evolution, also requires $5$ samples to achieve the desired effect. Based on empirical observations, a good strategy for initial experiments is to choose the number of samples to be approximately equal to the square root of the number of degrees of freedom, that is, $n \approx \sqrt{d}$. Indeed, by implementing the sampling algorithm with $60$ samples, we are able to achieve a numerical performance that nearly reproduces the results obtained by the baseline adjoint method for the optimal precursors. However, it has been observed that the numerical results are unstable. Out of four runs, only one consistently produces correct numerical performance."

**1.2 Typos**

1. *line 37: "is" → "was"*
   Thanks, we have revised it.

2. *line 67: "reivew"→ "review"*
   Thanks, we have revised it.

3. *page 3 footnote: "constrained optimization"*
   Thanks, we have revised it.

4. *line 94: "furthre researches" → "further research"*
   Thanks, we have revised it.

5. *line 252: "probably" → "probable"*
   Thanks, we have revised it.

**2 The Second Reviewer's Report**

**2.1 Major Comments**

1. *In the Introduction (Lines 50-55), the authors state that "there is a great similarity between optimal precursors and the optimal initial errors obtained by CNOP approaches, in terms of spatial structure and localization." The readers would appreciate a few sentences as to why this is the case, since it is not, in my opinion, trivial. Are the optimal initial errors the predictors for ENSO in the ZC model?*
   Thanks for the good comment. Following your suggestions, we have revised them in two places.

   (1) Regarding the optimal precursors, we provide the explanation and the reference in Line $50 -$ Line $52$ as " **In the study (Duan et al., 2004), it was found that the CNOP approach using ZC-specified climatology with the seasonal cycle as the basic state produces optimal initial errors, which act as the optimal precursors for triggering ENSO events.**"

(2) For the optimal growth errors: we also provide the explanation and the reference to show its similarity with the optimal precursors for the ENSO event in Line 58 – Line 60 as "**The study by Mu et al.(2014) verified that the optimal precursors obtained in the ZC model exhibit significant similarity with the optimal initial growth errors, which are obtained by considering the ENSO events triggered the optimal precursors as a basic state.**"

2. *In Lines 64-70, the authors speak about optimal energy growth. Typically optimal energy is attained when the system (say, Navier-Stokes) is initialized with the optimal mode. Can the authors say something about the formal mathematical connection between these optimal modes and the adjoint methods? Could they potentially have the same spatial structure as other CNOP approaches? Is there literature that applies the optimal growth theory with ENSO?*

Thanks for the good comments. Following your suggestions, we have revised them in two places.

- Regarding the connection between these optimal modes and the adjoint methods and the discussion about the different optimization algorithms to obtain the CNOPs, we add a detailed description to Footnote 2 of Page 2 as "**It is worth noting that the first-order optimization method employed to obtain the maximum in the scientific community of fluid mechanics is the method of Lagrange multipliers (Kerswell, 2018), which has shown the consistent results when compared to other first-order optimization method mentioned in the following paragraph.** The method of Lagrange multipliers is a classical method to solve the **constrained** optimization problem. It involves transforming the constrained optimization problem into an unconstrained one by incorporating the constrained condition into the Lagrange multipliers (Nocedall and Wright, Chapter 12). **Additionally, the adjoint method is also explored to numerically compute the gradient. The details of the solution procedure can be found in (Kerswell, 2018).**"

- Yes, there is some literature that applies to the optimal growth theory with ENSO, but it is not about the optimal energy growth theory. Almost of the literature is listed in Line 50 – Line 65 as "**(Duan et al., 2004), (Mu et al., 2007), (Duan et al., 2008), (Yu et al., 2009), (Mu et al., 2014), (Duan et al., 2014), (Tao et al.,2020), and (Zheng et al.,2023).**

3. *Regarding the optimal energy growth, it might be useful to cite the seminal work on linear energy growth by [Reddy and Henningson, 1993].*

Thanks, we have cited the seminal work [Reddy and Henningson, 1993] and revised the sentences in Line 66 to Line 75 as " **Within the field of fluid mechanics, turbulence is widely regarded as a crucial and highly influential topic. The study of optimal energy growth was initially explored using the non-normal mode method in the seminal work by Reddy and Henningson [1993]. Additionally, the scientific community has also developed the CNOP approach to investigate the disturbance of least amplitude for transition to turbulence. The CNOP approach, as described**

in (Pringle and Kerswell, 2010; Cherubini et al., 2010; Monokrousos et al., 2011), identifies the optimal precursors, referred to as minimal seeds, for the transition to turbulence. Nonlinear nonmodal analysis is applied to the 3D Navier-Stokes equation for an incompressible fluid to determine the optimal energy growth over all disturbances with a given starting energy and time horizon (Pringle and Kerswell, 2010; Cherubini et al., 2010; Monokrousos et al., 2011). Further details can be found in the comprehensive review (Kerswell, 2018) and an earlier review (Kerswell et al.,2014)."

4. *Line 114: can the authors explain the choice of $\tau = 9$?*

The prediction times $\tau$ were selected based on the four seasons of a year, specifically 3, 6, 9, and 12 months. However, a season of $\tau = 3$ months was deemed too short for effective prediction. Therefore, numerical experiments were conducted for $\tau = 6$, $\tau = 9$, and $\tau = 12$ months. The spatial patterns of SSTA and thermocline are almost consistent across all three cases, so we choose the middle case, $\tau = 9$ months, as a representative shown in the paper. The subsequent spatial patterns correspond to the cases with prediction times of $\tau = 6$ months and $\tau = 9$ months, respectively.

(1) The prediction time is 6 months.

[Figure]

(a) SST Anomalies  (b) Thermocline Depth Anomalies

Figure 1: The prediction time is 6 months.

(2) The prediction time is 12 months.

[Figure]

(a) SST Anomalies                    (b) Thermocline Depth Anomalies

Figure 2: The prediction time is 12 months.

5. *Line 114: can the CNOP's spatial structure change as a function of $\tau$?*

   Thanks, this is a very good question. In theory, the spatial structure of the CNOP should be a function of $\tau$. However, the ZC model is developed to describe the main physical mechanism of the ENSO events, Bjerknes positive feedback in the equatorial oceans. In other words, for different $\tau$s, they share the same mechanism in the ZC model. Hence, the spatial structures of the CNOPs obtained from the settings of different $\tau$ are almost consistent.

6. *I understand the Section 2.2 is a brief summary of [Shi and Sun, 2023], although the reader would benefit from a couple extra words on the actual statistical machine learning algorithm employed. How is the objective function actually maximised? If evaluating $J(\boldsymbol{u}_0 + \epsilon \boldsymbol{v}_0)$ requires solving the equations to find out $\boldsymbol{T}(\tau)$, how can it be less costly than the adjoint method? Can the authors give an intuition?*

   Thanks for the comment. We add the sentences to describe how the objective function is actually maximized in Line 138 to Line 142 as "**By utilizing the sample average of function values (7) as an approximate gradient, we can employ various gradient accent methods within the constraint domain, such as SPG, SQP, BFGS and the**

method of Lagrange multiplier, which help us maximize the objective function $J(\boldsymbol{u_0})$. In this paper, the specific gradient accent method within the constraint domain that we utilize is the second spectral projected gradient (SPG2) method mentioned, as mentioned in (Birgin et al., 2000)." Additionally, we revised the following paragraph in Line 143 – Line 162 to provide an intuition and explain how it can be less costly than the adjoint method as "**The average of function values (7) indicates that the random variables $\boldsymbol{v}_{0,i}, (i = 1,\ldots,n)$ are independently sampled from the uniform distribution on the unit sphere $\mathbb{S}^{d-1}$. This means that for any two samples, $\boldsymbol{v}_{0,i}$ and $\boldsymbol{v}_{0,j}$, where the indices $i$ and $j$ satisfy $i \neq j$, there is no relationship between them. In other words, every sample $\boldsymbol{v}_{0,i}, (i \in \{1,\ldots,n\})$ has no influence with each other and is drawn independently. With modern parallel computation techniques, it is possible to run the numerical model and obtain the values $J(\boldsymbol{u}_0 + \epsilon\boldsymbol{v}_{0,i})\boldsymbol{v}_{0,i}$ for each $i \in \{1,\ldots,n\}$ simultaneously, assuming unlimited computational resources are available. This parallelization allows for efficient computation and reduces the time required to run $n$ instances of the numerical model to that of running the model only once. However, it is important to note that in the adjoint method, the process of running the numerical model involves two consecutive steps. Initially, there is a forward numerical integration from $0$ to $\tau$, followed by a backward numerical integration from $\tau$ to $0$. These computations are based on the data obtained by running the numerical model. This process is executed in a single-thread manner, meaning that parallel computation is not applicable. On the other hand, in the implementation of the sampling algorithm, the process of running the numerical model only requires forward numerical integration from $0$ to $\tau$, without any backward numerical integration. This implies that for each sample, we only need to run the forward numerical integration once. Since the samples are independent, we can leverage the parallel computation to implement the sampling algorithm, which further reduces the time required for running the forward numerical integration once. With the current resource of computation, we have successfully implemented the sampling algorithm to obtain the CNOPs, or the optimal precursors of the ZC model. This numerical model has a substantial number of degrees of freedom, estimated to be on the order of $O(10^4 - 10^5)$. The implementation utilizes the modern parallel computation technique. The numerical performance, including the spatial patterns, objective values, computation times and nonlinear evolution of Nino 3.4 index, is shown in Section 3.**"

7. *Line 235: For La Niña events, how would your algorithm change (in the language of Section 2.2)?*

Our algorithm follows a similar way as the adjoint method for the La Niña event, as depicted in Figure 3. The black line in Figure 3 shows exhibits a similar nonlinear evolution behavior to that in (Duan et al.,2008, Figure 5), with a little difference, where we use the current internationally recognized Niño 3.4 index instead of the previously used Niño 3 index. In Figure 3, the blue curve represents the nonlinear evolution of the Niño 3.4 index for the sampling algorithm with $n = 1000$, and the red one for the sampling algorithm with $n = 200$.

[Figure]

Figure 3: The nonlinear time evolution of Niño 3.4 SST anomaly index within a model year.

8. *For Figures 3 and 4, some sort of error bars would be useful. Could the authors possibly use the theory and bounds of [Shi and Sun, 2023] to compute some uncertainty bars?*
Thanks for the good suggestion.

   (1) We re-run the sampling method 50 times to obtain the optimal precursors. The error bars are added in Figures 3 and 4 to represent the range of errors.

   (2) Yes, the theory and bounds of [Shi and Sun, 2023] provide a kind of bar to measure the uncertainty of the gradient computed by the sampling method with a high probability (exponential tail), which is similar to the confidence interval in statistics (a bar covers 95%).

**2.2 MinorTechnical Comments**

1. *Line 2: I suggest using "tropical Pacific" instead of "tropic Pacific".*
Thanks, we have revised it.

2. *Line 51: It would be helpful to clarify what "spring predictability barrier" means and its causes.*
Thanks, we have added the means of "spring predictability barrier" and its causes immediately following the sentence commented in Line 53 – Line 56 as **The SPB refers to a phenomenon in climate science where the predictability of systems, such as El Niño or La Niña, significantly decreases during the spring season. This is likely due to the transitional nature of spring for ENSO, where signals are lweak and noise is high, making predictions more challenging.**

3. *Line 60: I suggest to use SPB instead of "spring predictability barrier".*
   Thanks, we have revised it.

4. *Line 61: remove "the" in "the turbulence".*
   Thanks, we have revised it.

5. *Line 67: replace "reivew" by "review".*
   Thanks, we have revised it.

6. *Line 94: replace "futhre" by "further".*
   Thanks, we have revised it.

7. *Lines 111-112: awkward phrasing: "Since the target quantity required to maximize that we concern...".*
   Thanks, we have revised the sentence as "**As our primary concern is maximizing the target quantity solely dependent on the nonlinear evolution state of the SST anomalies, we define the objective function as:**".

8. *Eq. (6): "$\mathbb{S}^{d-1}$" is not defined until way later. I would suggest defining it immediately below.*
   Thanks, we add the definition of "$\mathbb{S}^{d-1}$" following your suggestion.

9. *Line 237: replace "such" by "so".*
   Thanks, we have revised it.

**References**

S. C. Reddy and D. S. Henningson. Energy growth in viscous channel flows. *Journal of Fluid Mechanics*, 252:209–238, 1993.

B. Shi and G. Sun. An adjoint-free algorithm for conditional nonlinear optimal perturbations (cnops) via sampling. *Nonlinear Processes in Geophysics*, 30(3):263–276, 2023.